# Effect of Vitamin D Supplementation on the Cerebral Placental Ratio in Pregnancy Complicated with Early Fetal Growth Restriction

**DOI:** 10.3390/jcm11092627

**Published:** 2022-05-07

**Authors:** Karolina Jakubiec-Wisniewska, Hubert Huras, Magdalena Kolak

**Affiliations:** Department of Obstetrics and Perinatology, Jagiellonian University Medical College, 31-501 Krakow, Poland; karolina.jakubiec@uj.edu.pl (K.J.-W.); hubert.huras@uj.edu.pl (H.H.)

**Keywords:** fetal growth restriction (FGR), vitamin D, cerebral placental ratio (CPR)

## Abstract

Fetal growth restriction (FGR) is a complication of pregnancy connected with increased risk of intrauterine fetal demise. To increase the diagnostic accuracy, the cerebral placental ratio (CPR) is used. Vitamin D may play a role in the regulation of vascular flow in the fetus. The aim is to assess the relationship between CPR and vitamin D supplementation in fetuses with early FGR. It is a prospective cohort study. Pregnant females were divided into groups with 2000 IU and <500 IU of vitamin D. Both groups were observed for 14 days; USG was performed three times with one-week intervals. EFW and CPR were measured. Absolute CPR values were initially observed to differ significantly (*p* = 0.0032). Measurements on the seventh day of observation indicated that CPR was significantly higher (*p* = 0.0455) in fetuses of patients receiving vitamin D at a dose of 2000 IU 1.75 (IQR: 1.47; 2.06) vs. <500 IU group 1.55 (IQR: 1.04; 1.52). Similarly, on day 14: (*p* < 0.0001)—2.39 (IQR: 1.82; 2.69) vs. 1.21 (IQR: 0.98; 1.52). Supplementation with vitamin D at a dose of 2000 IU may have an influence on the increase in the CPR in fetuses with early FGR.

## 1. Introduction

Fetal growth restriction (FGR) is defined as growing below the genetically expected potential [1]. We may suspect FGR when two measurements of the fetus taken during pregnancy show too slow growth rate and/or when the fetus does not reach the expected estimated weight in relation to the gestational age [2]. It is believed that FGR is a consequence of disturbances in the hemostasis of the feto-placental unit and placental insufficiency [3]. Currently, the causes of FGR can be divided into maternal (e.g., maternal diseases: hypertension, diabetes; drugs used during pregnancy; smoking; nutritional deficiencies, underweight, overweight), related to obstetric history (e.g., giving birth to a small child in a previous pregnancy, short or long intervals between pregnancies), associated with the course of the current pregnancy (e.g., pre-eclampsia, pregnancy-induced hypertension, heavy bleeding) and paternal (father’s birth weight below the 10th percentile). These factors can be divided into large and small depending on the OR (odds ratio), AOR (adjusted odds ratio) or RR (relative risk). The diagnosis of an increased risk of FGR is possible only when one large factor or three small factors are found [4,5,6,7,8]. Still, in about 40% of cases of fetal growth restriction, it is not possible to identify its cause [9]. Previously, maternal, fetal (e.g., chromosomal aberrations, birth defects), and placental risk factors (e.g., incorrect placement of the placenta, detachment of the placenta, hematoma, incorrect location of the umbilical cord, abnormal structure of the placenta) were distinguished [10,11]. However, due to the currently adopted definition of fetal growth restriction, according to which the diagnosis of FGR is made after excluding congenital abnormalities, chromosomal aberrations, and TORCH infections, the former classification of risk factors for a long time is not applicable at present.

Since the vast majority of FGR cases are caused by placental insufficiency, the evaluation of this organ’s function using the Doppler examination of the umbilical artery is the standard of clinical practice for differentiating between FGR and SGA [12,13,14,15]. When FGR is suspected, routine assessment should be made of blood flow in the umbilical artery (UA) as well as in the fetal middle cerebral artery (MCA) [16,17].

In order to increase the diagnostic accuracy, i.e., a more precise assessment of blood flow redistribution, the so-called cerebral placental ratio (CPR) is used; this ratio is the quotient between the flow rates in the middle cerebral artery and the flow rates in the umbilical artery. The limit value is 1.08, above which the flow velocity is considered normal [18,19].

Vitamin D may play a role in the regulation of vascular flow in the fetus by affecting the blood vessels in the placenta. In a study on mice, Liu et al. and Tesic et al. proved that in mice restricted to a diet of vitamin D, there was a reduced diameter of the vessels and a reduced length and volume of capillaries in the placenta, especially in the zone responsible for the exchange of nutrients [20,21]. Grundmann et al. also confirmed that vitamin D promotes the formation of capillary structures and the upregulation of an angiogenic factor—vascular endothelial growth factor (VEGF) [22,23]. Similar relationships were found in studies of pregnant women. Cell culture studies have shown that vitamin D influences the angiogenesis process in the placenta [24]. The dependence of vitamin D not only with VEGF but also with anti-angiogenic factors has been demonstrated. A randomized controlled trial with 43 patients who took 400 IU or 4400 IU Vitamin D3 daily showed a significant relationship between the dose taken and maternal vitamin D status and the expression of soluble fms-like tyrosine kinase 1 (sFlt-1) at mRNA level. The decreased level of vitamin D increased the risk of spontaneous abortion [25].

Normal serum levels of vitamin D and supplementation were affected on the appropriate transcription of genes in the placenta and favored the reduction of anti-angiogenic factors, and thus, the improvement of placental angiogenesis [26]. In the available scientific literature, publications are available reporting that targeted management of expansion of the volume of the vascular bed in the maternal–fetal circulation translates into the improvement of peripheral flows in the fetus. In a study involving patients with abnormal flow in the umbilical artery in the fetus (AEDV or REDV), dilation of placental blood vessels with l-arginine and pravastatin led to a significant improvement in umbilical cord flow in the fetus, which allowed for further continuation of pregnancy [27]. It can be concluded that since vitamin D has a multidirectional effect on the improvement of placental angiogenesis, it will also affect the improvement of peripheral flows in the fetus. The aim of the study is to assess the relationship between CPR and vitamin D supplementation in fetuses with early FGR.

## 2. Materials and Methods

The study is a prospective cohort study that took place at the Department of Obstetrics and Perinatology of the Jagiellonian University Medical College, between October 2017 to March 2019. It is a third reference center to which patients with complications of pregnancy are referred. A total of 100 patients with early FGR were followed. The study was carried out from October to March inclusive, because at that time there was no skin synthesis of vitamin D that could affect the results of the measurements. The group was divided into two subgroups according to vitamin D intake: to be optimal and suboptimal. The division into subgroups was based on the patient’s history of vitamin D intake, i.e., information from the questionnaire completed by the patient. The patient did not need to know the dose of vitamin D taken; they provided the name of the vitamin preparation they were taking, as well as the frequency and regularity of supplementation. The dose of vitamin D in a given preparation was determined on the basis of the online Drug Index. The first (optimal) research group (*n* = 50) were patients who were taking vitamin D at the dose of 2000 IU recommended in the Recommendation of the Polish Gynecological Society for vitamin D supplementation (*n* = 50). The second (suboptimal) were patients who did not take vitamin D at all or supplemented vitamin D in the dose found in most vitamin preparations for pregnant women, i.e., below 500 IU. Age in both groups did not differ significantly. In the optimal group, the second quartile was 29.00 years, interquartile range (27.75; 35.25); in suboptimal group, the second quartile was 28.00 years, interquartile range (25.50; 32.00), *p* = 0.0580.

Inclusion criteria were as follows: early FGR, single pregnancy, duration of pregnancy before 32 weeks, age 18–45 years. The study excluded patients with diabetes, hypertension, hypothyroidism (FGR risk factors), following a vegetarian/vegan diet, and smoking. An additional exclusion criterion was the presence of an unambiguous cause, which is known to inhibit the fetal growth potential, e.g., previously diagnosed genetic disorders in the fetus. The study was approved by the Bioethics Committee, no. 122.6120.262.2016, on 29 September 2016.

FGR was determined according to the Delphi criteria. The study included patients whose EFW was below the 10th percentile and who had UA PI above the 95th percentile, or the EFW was below the 3rd percentile, or had AEDF in the umbilical artery. All patients gave their written consent to participate in the study. There was no clinical intervention in the study. The patients were followed for 14 days. The clinical part of the study included an ultrasound examination (Samsung HS60, Seoul, Korea). They were performed three times, with an interval of 1 week. The test consisted of a standard measurement of fetal biometry. In addition, the umbilical artery (UA PI) and the middle cerebral artery flows were assessed and CPR was calculated. The ultrasound examinations were performed according to local and international guidelines [28,29].

### Statistics

The data were collected, analyzed, and processed in the STATISTICA program. The results are summarized in tables and presented in diagrams. Nominal variables are summarized as counts and percentages. Depending on the distribution of variables, the traits with continuous characteristics were presented as means with standard deviation (SD) or as medians with interquartile range (IQR). Nominal variables were compared using the Pearson chi-square test or the exact Fischer test, when more than 20% of cells had expected numbers lower than 5; in the case of several-variant variables measured on an ordinal scale (e.g., pregnancy number), the Cochran–Armitage test was used and for the trend. In the case when a given variable took many values, the Mann–Whitney test was applied. The normality of the distributions in the groups was assessed using the Shapiro–Wilk test, while the homoscedasticity was calculated using the Leven test. Differences between continuous variables with a normal distribution when comparing the two groups were tested using the Student’s *t*-test or the Welch’s *t*-test, depending on the assumption of homoscedasticity. In the case of variables with a distribution significantly different from the normal one, the Mann–Whitney test was used. In order to investigate the relationship between continuous variables, the study used the Pearson’s linear correlation coefficient (for variables with a normal distribution) and the Spearman’s rank correlation coefficient, which speaks of a monotonic dependence. In order to investigate the multivariate influence of variables on selected continuous parameters, the methods of linear regression were used. The standard significance level for all analyses is 0.05. All hypotheses are two-sided [30].

## 3. Results

At the beginning of the study, vitamin D level in both group were checked. In suboptimal group, it was significantly lower at 23.04 ng/mL, interquartile range (16.03; 29.25), whereas in optimal group it was 36.18 ng/mL, interquartile range (26.48; 45.26), *p* < 0.0001.

A total of 100 patients (50 in optimal group and 50 in suboptimal group) participated in the study. During 14 days of observations, eight patients in suboptimal group had indication of requiring cesarean delivery because of signs of fetal decompensations (all eight cases fell out of observation between 8 and 14 days of study).

In patients supplementing vitamin D in a dose below 500 IU compared with those taking a dose of 2000 IU, abnormal CPR was significantly more frequent at the time of study entry (*p* = 0.0050) at 36% vs. 12%; as well as abnormal CPR after 7 days of observation (*p* = 0.0014) at 40% vs. 12%; and after 14 days of observation (*p* < 0.001) at 47.62% vs. 8% (Table 1).

Absolute CPR values were initially observed in fetuses of patients receiving vitamin D at a dose of 2000 IU; those taking vitamin D in a dose below 500 IU differed significantly (*p* = 0.0032). The medians were 1.90 (IQR: 1.54; 2.32) and 1.47 (IQR: 1.16; 2.02), respectively. Similarly, significant (*p* = 0.0181) results were observed for the CPR percentiles 33.00 (IQR: 12.25; 66.25) and 11.00 (IQR: 2.75; 40.50), respectively. Measurements on the seventh day of observation indicated that CPR was significantly higher (*p* = 0.0455) in fetuses of patients receiving vitamin D at a dose of 2000 IU, 1.75 (IQR: 1.47; 2.06), than in fetuses of patients taking vitamin D below 500 IU, 1.55 (IQR: 1.04; 1.52). Similarly, differences were observed on day 14 of observation (*p* < 0.0001)—2.39 (IQR: 1.82; 2.69) vs. 1.21 (IQR: 0.98; 1.52). Higher values for the CPR percentile were also observed in fetuses of patients taking vitamin D at a dose of 2000 IU but only on day 14 (*p* < 0.0001): 69.00 (IQR: 25.27; 92.25) vs. 5.00 (IQR: 1.00; 13.50), which was not observed on day 7 (*p* = 0.0723); 22.00 (IQR: 9.00; 46.75) vs. 11.00 (IQR: 1.75; 36.25) (Table 2).

Differences in the dynamics of changes for absolute CPR values were also observed in the fetuses of patients receiving vitamin D at a dose of 2000 IU and those taking vitamin D at a dose below 500 IU, but only on the 14th day. The difference between the value measured on day 14 of observation and the baseline value was lower in the group of patients receiving vitamin D at a dose of 2000 IU, respectively: 0.23 (IQR: −0.18; 0.64) vs. −0.33 (IQR: −0.64; −0.11), *p* < 0.0001. There was no difference in this parameter between day 7 and baseline (*p* = 0.7933). A significant difference was also observed for changes in CPR percentile between day 14 and baseline (*p* < 0.0001). The difference between the percentile CPR value measured on the 14th day of observation and the baseline value was lower in the group of patients receiving vitamin D at a dose of 2000 IU, respectively: 3.00 (IQR: −6.75; 30.25) vs. −9.00 (IQR: −30.75; −1.00), *p* < 0.0001. Similarly, no difference was observed between the percentile CPR value measured on the 7th day of observation (*p* = 0.8683) (Figure 1, Figure 2 and Figure 3).

A multivariate analysis was performed: maternal age, gestation week, number of pregnancies, and vitamin D dose were included in the CPR prediction models. D dose below 500 IU was lower on average by 0.3994 (95% CI: −0.7218; −0.077) than in the group of women taking vitamin D at a dose of 2000 IU (*p* = 0.0157). However, it did not depend on the weeks of pregnancy (*p* = 0.7057).

CPR measured on the 7th day of observation was not significantly dependent on the dose of vitamin D intake (*p* = 0.1265) or the week of pregnancy (*p* = 0.7145).

CPR measured on day 14 of observation in women taking vitamin D doses below 500 IU was lower on average by 0.9042 (95% CI: −1.1748; −0.6332) than in the group of women receiving vitamin D dose of 2000 IU (*p* < 0.0001) assuming fixed values of other factors (maternal age, week of pregnancy, and number of previous pregnancies).

The decrease in CPR in patients who took vitamin D at a dose below 500 I was also significantly higher after 2 weeks of observation than in patients who took vitamin D at a dose of 2000 IU, respectively: 0.5708 (95% CI: −0.8186; −0.3232, *p* < 0.0001) assuming fixed values of other factors (maternal age, gestation week, and number of pregnancies). However, after one week of observation, it was not statistically significant (*p* = 0.1312) (Table 3).

There was a significant positive relationship between CPR and CPR percentile and vitamin D level on the 7th day of observation, r = 0.3210 (*p* = 0.0011) and r = 0.3310 (*p* = 0.0008), and also on the 14th day of observation, r = 0.2569 (*p* = 0.0134) and r = 0.2380 (*p* = 0.0223). There was no significant correlation between vitamin D levels and MCA PI and MCA PI percentiles at study entry, as well as the difference (between day 7 and day 14 and baseline) of MCA PI and MCA PI percentiles (Table 4, Figure 4 and Figure 5).

## 4. Discussion

Neilson et al. demonstrated that monitoring of fetal wellbeing with Doppler techniques reduced the incidence of perinatal mortality from high-risk pregnancies in 35% of cases [31]. Monitoring of UA flows reduces the risk of obstetric complications in pregnancies with suspected FGR [32]. A meta-analysis of nearly 7000 high-risk pregnancies showed that the assessment of umbilical artery flow significantly reduced the number of induction of labor and cesarean sections, and the percentage of perinatal deaths [32]. Abnormal flow in UA is associated with increased perinatal mortality (26% vs. 6%), perinatal complications in neonates and developmental disorders in the nervous system observed in children (35% vs. 0%) [33]. The scientific literature reports that the occurrence of flow disorders in the umbilical artery of the nature of atrophy or retrograde end-diastolic flow is preceded by an acute deterioration of the fetal condition by an average of 7 days. Their occurrence should be an indication for earlier termination of pregnancy, depending on the duration of pregnancy, respectively [34].

A study by Cosmi et al. showed pathological dilatation of this vessel in 80% of fetuses 14 days before their acute health deterioration [35]. According to Gramellini, the increase in diastolic flow observed in FGR in MCA is a symptom of the beginning of circulatory centralization, i.e., an increase in blood flow through the brain as a compensation for decreased placental perfusion [36]. MCA diastolic causes an increase in the diastolic flow velocity and a decrease in the pulsation index, which is associated with an unfavorable prognosis, regardless of the flow in the umbilical artery [35]. The disappearance of the circulatory centralization phenomenon, despite persistent unfavorable conditions for the fetus, proves the decompensation of autoregulatory mechanisms. This phenomenon precedes the threat of intrauterine death and usually coexists with cerebral edema developing as a result of hypoxia [37].

In most cases, the cause of FGR is placental insufficiency, and its condition is reflected in the flow in the umbilical artery. The widening of the vascular bed of the maternal–fetal circulation improves the flow in the umbilical artery. On the other hand, hypoxia at the placental level is associated with a decrease in end-diastolic flow in the umbilical artery, which is manifested by an increase in the pulsation index in UA.

In the case of fetal growth restriction, especially in the later weeks of pregnancy, the flows in the umbilical artery and the middle cerebral artery often fluctuate within the normal range. In these cases, the usefulness of CPR determination, which is the quotient of the pulsation indices UA and MCA, is postulated. This coefficient may assume an incorrect value even when the flows in UA and MCA assessed separately are normal [38]. At study entry, 24% of patients had abnormal CPR, although abnormal UA PI and MCA PI occurred in only 10% and 14% of pregnant women, respectively. During the follow-up, abnormal CPR was found in 26% of patients who survived to the end of the follow-up. Taking into account the fact that eight patients dropped out of the observation due to a significant deterioration in blood flow, abnormal cardiotocography, or death of pregnancy. Abnormal CPR index at the end of the study, i.e., after 2 weeks of observation, could be expected in almost 1/3 of patients. The cerebrospinal cord index is more sensitive and specific in detecting fetal hypoxia than the pulse rates measured separately in the umbilical artery and the middle cerebral artery [39]. Thanks to its use in the study group, vascular flow abnormalities were found; thus, the risk of hypoxia in more than twice as many fetuses than after using UA PI and MCA PI separately.

With the duration of pregnancy up to and including week 34, the physiological CPR value increases, after which time it should physiologically decrease slightly [40].

The abnormal CPR index was significantly more frequent at any time of observation in patients taking low doses of vitamin D—36 vs. 12% at study enrolment, 40 vs. 12% after one week, and 47.62 vs. 8% after two weeks of observation. It was found that CPR after both one week and two weeks was higher with relatively higher serum vitamin D concentrations. Assuming the established values of other factors (maternal age, gestational week, and number of previous pregnancies), it was found that at the time of study entry in women taking vitamin D at a dose below 500 IU, CPR was lower on average by 0.3994. Similarly, on the 14th day, it was lower by an average of 0.9042 than in the group of women taking vitamin D at a dose of 2000 IU.

Differences in the dynamics of CPR changes depending on the supplemented dose were also observed, but only on the 14th day of the study, where in the fetuses of patients taking vitamin D at a dose of 2000 IU, CPR increased slightly, while in the case of supplementation with vitamin D in low doses, it was significantly reduced by an average of 0.33, which after conversion to percentiles gave a reduction by 9 percentiles. At study entry, in the subgroup below 500 IU Vitamin D, it was abnormal in 36% of patients, while on day 14, in was in as many as 47.62% of patients.

The only currently known method of FGR treatment is delivery; therefore, the greatest attention should be paid to determining the appropriate time of delivery and assessing the risk of iatrogenic complications in comparison with exposure to intrauterine development conditions unfavorable for the child. In the TRUFFLE study (Trial of Randomized and Umbilical Fetal Flow in Europe), the authors suggest considering earlier delivery due to the higher number of neurological complications in children born after 40 weeks [41]. Unfortunately, there are still no clear time guidelines and how to terminate a complicated FGR pregnancy. The most important factors influencing the decision about the time of delivery are the gestational age and the flow in the umbilical artery: CPR< 1, UA PI > 95 centile, MCA PI < 5 centile (>32 wog), UtA PI > 95 centile, according to both the RCOG (Royal College of Obstetricians and Gynaecologists) and SOGC (Society of Obstetricians and Gynaecologists of Canada), allows for the postponement of delivery beyond the 37th week of pregnancy. This is also consistent with the Polish Society of Gynecologists and Obstetricians, which recommends that in pregnancies complicated with FGR without symptoms of threat to the life of the fetus, induction of labor should be performed after the age of 37 weeks due to an increased risk of intrauterine fetal death [42]. Preterm delivery of pregnancy depends on the gestational age and the severity of vascular changes [12,13,14,29,43].

## 5. Conclusions

In conclusion, the results of the above study prove the positive importance of vitamin D supplementation and the effect of higher serum vitamin D concentrations on the growth of the fetus with intrauterine growth restriction and on the improvement of peripheral blood flow in the fetus, which is synonymous with improvement of its wellbeing. Due to the lack of proven methods of treating FGR, the report on the positive supportive effect of any substance should not be omitted. Vitamin D supplementation in the recommended dose can only bring benefits, with the potential absence of side effects. Therefore, it seems appropriate to place special emphasis on supplementation with vitamin D in the recommended dose of 2000 IU in women with a complicated or at-risk FGR pregnancy, and in the case of abnormal serum vitamin D levels, to compensate for its level with even higher doses. The difference in CPR value alterations between the two groups is more significant than the absolute values. Day 14 demonstrates a benefit in values. Supplementation with vitamin D at a dose of 2000 IU may have an influence on the increase of the CPR in fetuses with early FGR.

## Figures and Tables

**Figure 1 jcm-11-02627-f001:**
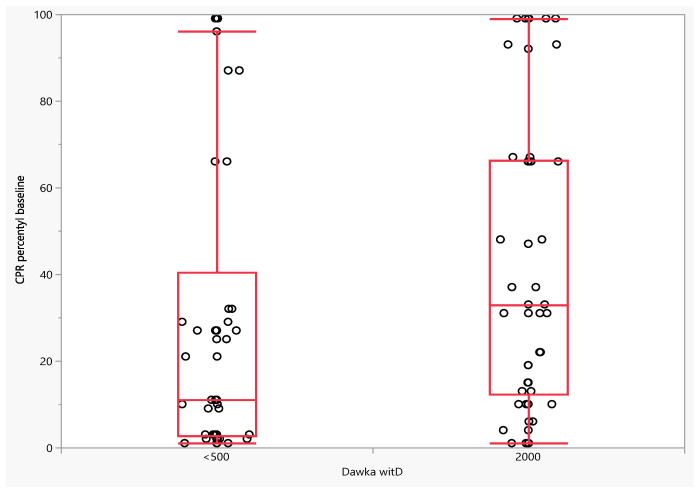
Percentile CPR at study entry depending on the dose of vitamin D ingested.

**Figure 2 jcm-11-02627-f002:**
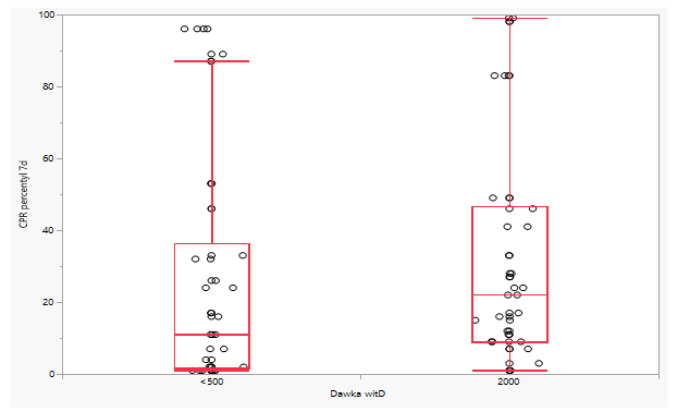
Percentile CPR on day 7 of observation depending on the dose of vitamin D.

**Figure 3 jcm-11-02627-f003:**
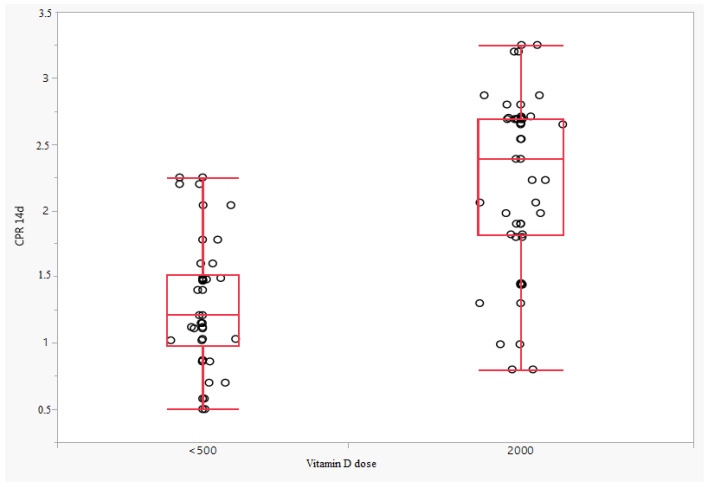
Percentile CPR on day 14 of observation depending on the dose of vitamin D.

**Figure 4 jcm-11-02627-f004:**
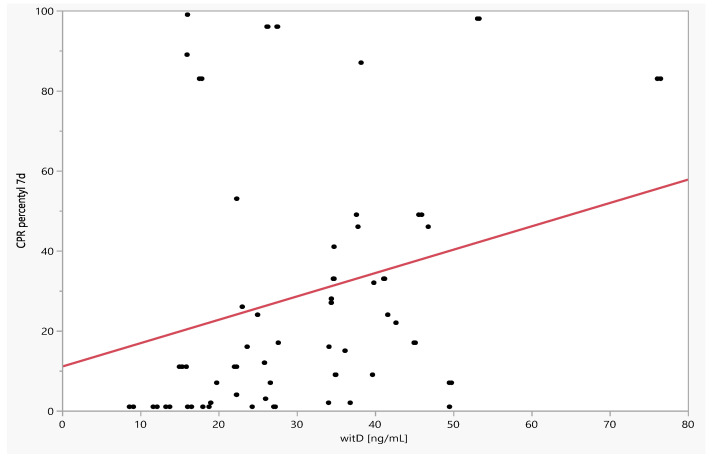
Dependence of CPR percentiles on day 7 of observation on the level of vitamin D.

**Figure 5 jcm-11-02627-f005:**
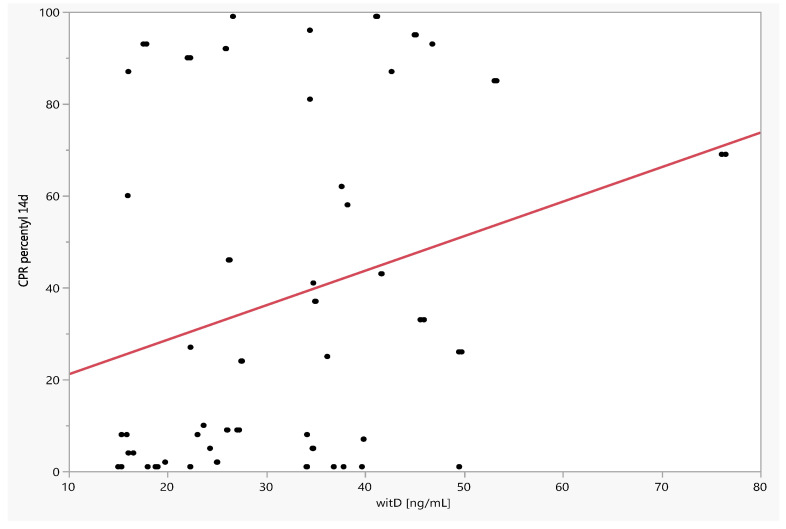
Dependence of CPR percentiles on day 14 of observation on the level of vitamin D.

**Table 1 jcm-11-02627-t001:** Comparison of the amount of abnormal cerebral umbilical cord ratio (CPR) in fetuses in patients supplementing vitamin D at the dose of 2000 IU recommended by the Polish Gynecological Society and vitamin D in a dose found in most vitamin preparations for pregnant women, i.e., below 500 IU.

Variable	Measure/Category	<500	2000	Total	*p*
Abnormal CPR percentile baseline	*n*	50	50	100	**0.0050**
	Yes	18 (36.00%)	6 (12.00%)	24 (24.00%)	
	No	32 (64.00%)	44 (88.00%)	76 (76.00%)	
Abnormal CPR percentile 7 d	*n*	50	50	100	**0.0014**
	Yes	20 (40.00%)	6 (12.00%)	26 (26.00%)	
	No	30 (60.00%)	44 (88.00%)	74 (74.00%)	
Abnormal CPR percentile 14 d	*n*	42	50	92	**<0.0001**
	Yes	20 (47.62%)	4 (8.00%)	24 (26.09%)	
	No	22 (52.38%)	46 (92.00%)	68 (73.91%)	

**Table 2 jcm-11-02627-t002:** Comparison of the cerebrospinal cord ratio (CPR) of fetuses in patients supplementing vitamin D at the dose of 2000 IU recommended by the Polish Gynecological Society and vitamin D in a dose found in most vitamin preparations for pregnant women, i.e., below 500 IU.

Variable	Measure/Category	<500	2000	Total	*p*
CPR baseline	*n*	50	50	100	**0.0032**
	Mean (±SD)	1.69 (±0.79)	2.01 (±0.69)	1.85 (±0.76)	
	Mean 95% CI	(1.47; 1.91)	(1.81; 2.21)	(1.70; 2.00)	
	Me (Q1; Q3)	1.47 (1.16; 2.02)	1.90 (1.54;2.32)	1.66 (1.33; 2.31)	
	Min/Max	0.70/3.50	0.60/3.75	0.60/3.75	
CPR 7d	*n*	50	50	100	**0.0455**
	Mean (±SD)	1.58 (±0.65)	1.80 (±0.59)	1.69 (±0.63)	
	Mean 95% CI	(1.39; 1.76)	(1.63; 1.97)	(1.57; 1.81)	
	Me (Q1; Q3)	1.55 (1.04; 1.92)	1.75 (1.47; 2.06)	1.64 (1.20; 2.01)	
	Min/Max	0.69/2.84	0.70/3.23	0.69/3.23	
CPR 14 d	*n*	42	50	92	**<0.0001**
	Mean (±SD)	1.30 (±0.49)	2.22 (±0.66)	1.80 (±0.74)	
	Mean 95% CI	(1.15; 1.45)	(2.03; 2.41)	(1.65; 1.95)	
	Me (Q1; Q3)	1.21 (0.98; 1.52)	2.39 (1.82; 2.69)	1.79 (1.15; 2.54)	
	Min/Max	0.50/2.25	0.80/3.25	0.50/3.25	
Odds CPR (7-d-baseline)	*n*	50	50	100	0.7933
	Mean (±SD)	−0.11 (±0.44)	−0.21 (±0.65)	−0.16 (±0.55)	
	Mean 95% CI	(−0.23; 0.01)	(−0.39; −0.02)	(−0.27; −0.05)	
	Me (Q1; Q3)	−0.01 (−0.40; 0.12)	−0.23 (−0.55; 0.27)	−0.01 (−0.46; 0.17)	
	Min/Max	−0.99/0.93	−1.70/1.11	−1.70/1.11	
Odds CPR (14-d-baseline)	*n*	42	50	92	**<0.0001**
	Mean (±SD)	−0.47 (±0.56)	0.21 (±0.56)	−0.10 (±0.65)	
	Mean 95% CI	(−0.65; −0.30)	(0.05; 0.37)	(−0.24; 0.04)	
	Me (Q1; Q3)	−0.33 (−0.64; −0.11)	0.23 (−0.18; 0.64)	−0.10 (−0.42; 0.35)	
	Min/Max	−1.90/0.42	−1.10/1.30	−1.90/1.30	
CPR percentile baseline	*n*	50	50	100	**0.0181**
	Mean (±SD)	30.64 (±36.10)	43.00 (±33.95)	36.82 (±35.41)	
	Mean 95% CI	(20.38; 40.90)	(33.35; 52.65)	(29.79; 43.85)	
	Me (Q1; Q3)	11.00 (2.75; 40.50)	33.00 (12.25; 66.25)	26.00 (6.00; 66.00)	
	Min/Max	1.00/99.00	1.00/99.00	1.00/99.00	
CPR percentile 7 d	*n*	50	50	100	0.0723
	Mean (±SD)	26.40 (±32.36)	31.60 (±29.82)	29.00 (±31.07)	
	Mean 95% CI	(17.20; 35.60)	(23.13; 40.07)	(22.84; 35.16)	
	Me (Q1; Q3)	11.00 (1.75; 36.25)	22.00 (9.00; 46.75)	16.50 (4.00;46.00)	
	Min/Max	1.00/96.00	1.00/99.00	1.00/99.00	
CPR percentile 14 d	n	42	50	92	**<0.0001**
	Mean (±SD)	13.19 (±18.67)	58.28 (±36.02)	37.70 (±36.95)	
	Mean 95% CI	(7.37; 19.01)	(48.04; 68.52)	(30.04; 45.35)	
	Me (Q1; Q3)	5.00 (1.00; 13.50)	69.00 (25.75; 92.25)	25.50 (4.00; 81.00)	
	Min/Max	1.00/60.00	1.00/99.00	1.00/99.00	
Odds CPR percentile (7-d-baseline)	*n*	50	50	100	0.8683
	Mean (±SD)	−4.24 (±18.74)	−11.40 (±36.21)	−7.82 (±28.91)	
	Mean 95% CI	(−9.56; 1.08)	(−21.69; −1.11)	(−13.56; −2.08)	
	Me (Q1; Q3)	−1.00 (−7.50; 0.00)	−1.00 (−33.75;13.00)	−1.00 (−19.00;6.00)	
	Min/Max	−55.00/44.00	−88.00/51.00	−88.00/51.00	
Odds CPR percentile (14-d-baseline)	*n*	42	50	92	**<0.0001**
	Mean (±SD)	−20.43 (±25.29)	15.28 (±29.24)	−1.02 (±32.69)	
	Mean 95% CI	(−28.31; −12.55)	(6.97; 23.59)	(−7.79; 5.75)	
	Me (Q1; Q3)	−9.00 (−30.75; −1.00)	3.00 (−6.75; 30.25)	−1.00 (−20.00; 20.00)	
	Min/Max	−75.00/3.00	−23.00/83.00	−75.00/83.00	

**Table 3 jcm-11-02627-t003:** Summary of linear regression results explaining selected CPR parameters.

Model/Responsible Variable	Parameter	(95% CI)	*p*-Value	Statistics	Value
Model: CPR baseline			0.0100	R˛	12.92%
Intercept	2.9624	(1.3863; 4.5385)	**0.0003**	Adjusted R˛	9.26%
Mother age (years)	−0.0471	(−0.0823; −0.0119)	**0.0092**	Observations	100
Gestational age (weeks)	0.0093	(−0.0394; 0.0579)	0.7057		
Pregnancy	0.0151	(−0.2017; 0.2318)	0.8906		
Dose of vitD (<500)	−0.1997	(−0.3609; −0.0385)	0.0157		
Model: CPR 7 d			0.1079	R˛	7.60%
Intercept	2.0236	(0.6784; 3.3687)	**0.0036**	Adjusted R˛	3.71%
Mother age (years)	−0.0132	(−0.0432; 0.0168)	0.3840	Observations	100
Gestational age (weeks)	0.0077	(−0.0338; 0.0492)	0.7145		
Pregnancy	−0.1133	(−0.2983; 0.0717)	0.2271		
Dose of vitD (<500)	−0.1068	(−0.2444; 0.0307)	0.1265		
Model: CPR 14 d			<0.0001	R˛	41.83%
Intercept	2.2803	(0.9709; 3.5896)	**0.0008**	Adjusted R˛	39.15%
Mother age (years)	0.0062	(−0.0248; 0.0373)	0.6899	Observations	92
Gestational age (weeks)	−0.0145	(−0.0556; 0.0266)	0.4849		
Pregnancy	−0.1792	(−0.3610; 0.0025)	0.0532		
Dose of vitD (<500)	−0.4521	(−0.5874; −0.3168)	**<0.0001**		
Model: Róznica CPR (7-d-baseline)			0.1227	R˛	7.29%
Intercept	−0.9388	(−2.1223; 0.2446)	0.1186	Adjusted R˛	3.38%
Mother age (years)	0.0339	(0.0075; 0.0603)	**0.0124**	Observations	100
Gestational age (weeks)	−0.0016	(−0.0381; 0.0349)	0.9305		
Pregnancy	−0.1284	(−0.2911; 0.0344)	0.1208		
Dose of vitD (<500)	0.0928	(−0.0282; 0.2139)	0.1312		
Model: Róznica CPR (14-d-baseline)			<0.0001	R˛	36.90%
Intercept	−0.6243	(−1.8224; 0.5737)	0.3032	Adjusted R˛	34.00%
Mother age (years)	0.0503	(0.0219; 0.0787)	**0.0007**	Observations	92
Gestational age (weeks)	−0.0234	(−0.0610; 0.0143)	0.2203		
Pregnancy	−0.2011	(−0.3674; −0.0348)	**0.0183**		
Dose of vitD (<500)	−0.2854	(−0.4093; −0.1616)	**<0.0001**		

**Table 4 jcm-11-02627-t004:** Spearman’s rank correlation between the value of vitamin D and other parameters.

By Variable	Spearman’s Correlation	Spearman’s Correlation *p*-Value
CPR percentile 7 d	0.3310	**0.0008**
CPR 7 d	0.3210	**0.0011**
CPR 14 d	0.2569	**0.0134**
CPR percentile 14 d	0.2380	**0.0223**
Odds CPR (14-d-baseline)	0.1863	0.0754
CPR baseline	0.1649	0.1011
Odds CPR percentile (14-d-baseline)	0.1587	0.1308
CPR percentile baseline	0.1463	0.1463
Odds CPR percentile (7-d-baseline)	0.1304	0.1960
Odds CPR (7-d-baseline)	0.0529	0.6009

## Data Availability

Data sharing not applicable.

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
