# Peer review of "Effect of Vitamin D Supplementation on the Cerebral Placental Ratio in Pregnancy Complicated with Early Fetal Growth Restriction"

_jcm, 2022, doi:10.3390/jcm11092627_

Round 1
Reviewer 1 Report
The manuscript (ID: cancers-1651045) entitled “Effect of vitamin D supplementation on the cerebral placental ratio in pregnancy complicated with early fetal growth restriction” by Dr. akubiec-Wisniewska and colleagues describes a prospective cohort study which evaluated the relationship between cerebral placental ratio and vitamin D supplementation in fetuses with early fetal growth restriction. Despite the work is interesting, it can be improved in both writing style, figures and notions described. In my opinion, the ms can be accepted following a major revision. I have several comments for improving the manuscript:
Major comments
1. Spaces between paragraphs can be removed for a better reading, please revise the entire text accordingly
2. Mean age and SD should be included in lines 80 and 82
3. A brief description of the fetal growth restriction etiopathogenesis should be included. For instance, previous evidence indicate a relationship between pathogenic infections and negative pregnancy outcome (DOI: 10.1002/jcp.26952 and DOI: 10.1159/000482008). Have data been reported on a implication of this such of infections on fetal growth restriction? If yes, these notions on pregnancy outcome and fetal growth restriction should be, at least, briefly mentioned
4. Lack of vit D can increase the risk of spontaneous abortion (10.1590/1414-431X20176527), this information/reference should be included
5. Supporting references should be included in the statistics paragraph
6. The quality of all figures should be improved. Several words are almost unreadable as being too small
7. Observation days should be included in the methods
Minor observations
Line 15 better “pregnant females” instead of patients
Line 42 refs Liu et al. and Grundmann et al. should be included at the end of the sentence.
Lines
Lines 274-284 these sentences should be included in the “5. Conclusions “ section
Lines 143, 145 and others it is unclear for the reviewer the presence of “*” following the p values
Author Response
The manuscript (ID: cancers-1651045) entitled “Effect of vitamin D supplementation on the cerebral placental ratio in pregnancy complicated with early fetal growth restriction” by Dr. Jakubiec-Wisniewska and colleagues describes a prospective cohort study which evaluated the relationship between cerebral placental ratio and vitamin D supplementation in fetuses with early fetal growth restriction. Despite the work is interesting, it can be improved in both writing style, figures and notions described. In my opinion, the ms can be accepted following a major revision. I have several comments for improving the manuscript:
Major comments
1. Spaces between paragraphs can be removed for a better reading, please revise the entire text accordingly
Corrected
2. Mean age and SD should be included in lines 80 and 82
Added
3. A brief description of the fetal growth restriction etiopathogenesis should be included. For instance, previous evidence indicate a relationship between pathogenic infections and negative pregnancy outcome (DOI: 10.1002/jcp.26952 and DOI: 10.1159/000482008). Have data been reported on a implication of this such of infections on fetal growth restriction? If yes, these notions on pregnancy outcome and fetal growth restriction should be, at least, briefly mentioned
Added
4. Lack of vit D can increase the risk of spontaneous abortion (10.1590/1414-431X20176527), this information/reference should be included
Added
5. Supporting references should be included in the statistics paragraph
Added
6. The quality of all figures should be improved. Several words are almost unreadable as being too small
Largered
7. Observation days should be included in the methods
Added
Minor observations
Line 15 better “pregnant females” instead of patients
changed
Line 42 refs Liu et al. and Grundmann et al. should be included at the end of the sentence.
Added
Lines 274-284 these sentences should be included in the “5. Conclusions “ section
changed
Lines 143, 145 and others it is unclear for the reviewer the presence of “*” following the p values
removed
Thank you for a revision!
Reviewer 2 Report
This is a very interesting study investigating the role of high dose vitamin D supplementation in pregnancy in improving the CPR values that reflect partially an improvement on growth restriction consequencies.
I have 3 comments and I think these thoughts should be included in the manuscript:
- It would be interesting if we knew the serum vitamin D storage levels for the two groups, so as to define if deficiency restoration or high dose supplementation is actually the action proposed.
- The difference in CPR value alterations between the two groups is more significant than the absolute values. Day 14 demonstrates a benefit in values.
- CPR is useful but rarely used to determine the severity and action in order to improve outcome.
It is also interesting to know for the missing cases on day 14 if were delivered and why.
Otherwise, this is a well-written, well-analysed and well-presented paper.
Author Response
This is a very interesting study investigating the role of high dose vitamin D supplementation in pregnancy in improving the CPR values that reflect partially an improvement on growth restriction consequencies.
I have 3 comments and I think these thoughts should be included in the manuscript:
- It would be interesting if we knew the serum vitamin D storage levels for the two groups, so as to define if deficiency restoration or high dose supplementation is actually the action proposed.
Added
- The difference in CPR value alterations between the two groups is more significant than the absolute values. Day 14 demonstrates a benefit in values.
Added
- CPR is useful but rarely used to determine the severity and action in order to improve outcome.
Usefulness CPR in timing of delivery was added in discussion
It is also interesting to know for the missing cases on day 14 if were delivered and why.
Added
Otherwise, this is a well-written, well-analysed and well-presented paper.p
Thank you !
Round 2
Reviewer 1 Report
The authors have satisfactorily addressed most of my concerns. The ms can be accepted in the present form